# Tolerogenic dendritic cell-based treatment for multiple sclerosis (MS): a harmonised study protocol for two phase I clinical trials comparing intradermal and intranodal cell administration

Barbara Willekens,[1,2] Silvia Presas-Rodríguez,[3,4] MJ Mansilla,[5,6] Judith Derdelinckx,[1,2] Wai-Ping Lee,[7] Griet Nijs,[7] Maxime De Laere,[2] Inez Wens,[2] Patrick Cras,[1] Paul Parizel,[8] Wim Van Hecke,[9] Annemie Ribbens,[9] Thibo Billiet,[9] Geert Adams,[10] Marie-Madeleine Couttenye,[11] Juan Navarro-Barriuso,[5,6] Aina Teniente-Serra,[5,6] Bibiana Quirant-Sánchez,[5,6] Ascensión Lopez-Diaz de Cerio,[12,13] Susana Inogés,[12,13] Felipe Prosper,[12,14] Anke Kip,[15] Herman Verheij,[15] Catharina C Gross,[16,17] Heinz Wiendl,[16,17] Marieke (SM) Van Ham,[18,19] Anja Ten Brinke,[18,19] Ana Maria Barriocanal,[20,21] Anna Massuet-Vilamajó,[22] Niel Hens,[23,24] Zwi Berneman,[2,7] Eva Martínez-Cáceres,[5] Nathalie Cools,[2,7] Cristina Ramo-Tello,[3] On behalf of the RESTORE consortium

BW and SP-R contributed equally.
NC and CR-T contributed equally.

NC and CR-T are senior authors. BW and SP-R are first authors.

For numbered affiliations see end of article.

**Correspondence to**
Nathalie Cools;
nathalie.cools@uza.be

Cristina Ramo-Tello;
cramot@gmail.com

## ABSTRACT

**Introduction** Based on the advances in the treatment of multiple sclerosis (MS), currently available disease-modifying treatments (DMT) have positively influenced the disease course of MS. However, the efficacy of DMT is highly variable and increasing treatment efficacy comes with a more severe risk profile. Hence, the unmet need for safer and more selective treatments remains. Specifically restoring immune tolerance towards myelin antigens may provide an attractive alternative. In this respect, antigen-specific tolerisation with autologous tolerogenic dendritic cells (tolDC) is a promising approach.

**Methods and analysis** Here, we will evaluate the clinical use of tolDC in a well-defined population of MS patients in two phase I clinical trials. In doing so, we aim to compare two ways of tolDC administration, namely intradermal and intranodal. The cells will be injected at consecutive intervals in three cohorts receiving incremental doses of tolDC, according to a best-of-five design. The primary objective is to assess the safety and feasibility of tolDC administration. For safety, the number of adverse events including MRI and clinical outcomes will be assessed. For feasibility, successful production of tolDC will be determined. Secondary endpoints include clinical and MRI outcome measures. The patients' immune profile will be assessed to find presumptive evidence for a tolerogenic effect in vivo.

**Ethics and dissemination** Ethics approval was obtained for the two phase I clinical trials. The results of the trials will be disseminated in a peer-reviewed journal, at scientific conferences and to patient associations.

**Trial registration numbers** NCT02618902 and NCT02903537; EudraCT numbers: 2015-002975-16 and 2015-003541-26.

## Strengths and limitations of this study

► The concept of the clinical trials is built on recent advances in the understanding of tolerance induction via cell-based therapy, and the effort made to precisely silence myelin-antigen-specific, putatively deleterious immune responses in the disease.

► The use of cryopreserved tolerogenic dendritic cells (tolDC) allows production of batches of tolDC, stored in ready-to-use aliquots for the required administrations during the treatment period, thereby reducing variability and global production cost.

► Harmonisation of clinical, MRI and immunological evaluations of the patients will enable us to compare results between two phase I clinical trials evaluating the safety and feasibility of autologous tolDC administration in patients with active MS.

► The patients are not randomised across the trials, limiting direct comparison of both routes of administration.

## INTRODUCTION

Multiple sclerosis (MS) is the leading cause of non-traumatic disability in young adults and typically presents between the age of 20–40 years, in the prime of a patient's personal and professional life. The heterogeneity of the disease course in MS, that is, relapsing-remitting or progressive, remains a challenge for patient management and design of clinical trials. According to Lublin *et al*,[1] MS is categorised as relapsing or progressive. In both

forms, disease activity (defined by clinical relapse and/or MRI lesions) and disease progression are measured. To date, more than 10 disease-modifying treatments (DMT) are approved for clinical use in patients with active relapsing-remitting MS (RRMS). This has dramatically influenced the management of MS patients in daily practice and has positively influenced prognosis of of a subset of MS patients. Indeed, whereas earlier epidemiological studies indicated that 50% of patients develop secondary progressive MS after 15 years when untreated,[2] only 18% of patients treated with modern MS therapeutics were reported to evolve to a secondary progressive course after a median duration of 16.8 years in a recent cohort.[3] Nonetheless, DMT are not specific nor selective for MS, that is, they work in an immunomodulatory or immunosuppressive way by sequestering or depleting lymphocytes, although that some therapies can induce immune reconstitution as well. Moreover, treatment-related side effects or risks can be severe,[4] leaving a significant and unmet need for safer and more disease-selective treatments.

The crux of MS is the patient's own immune cells attacking self-antigens in the central nervous system (CNS), caused by a loss of tolerance against myelin antigens. This is manifested by inflammatory infiltrates, demyelination and axonal loss, resulting in the clinical symptoms of the disease.[5 6] Previously, it was shown that proteins expressed in the myelin sheath, protecting neuronal axons of the CNS, are an important target of the autoreactive T cell response.[7–9] Also in our hands, ex vivo T cell reactivity against a mix of seven immunodominant myelin-derived peptides could be demonstrated in RRMS patients as compared with healthy controls and patients with other neurological disorders.[10] Although the underlying cause of the loss of tolerance towards myelin antigens has not been elucidated yet, one ultimate aim in the treatment of MS is to reestablish antigen-specific immune tolerance towards CNS structures.[11–15] In this perspective, it is postulated that tolerance-inducing antigen-specific therapy can be an innovative and promising strategy for the treatment of autoimmunity. This approach is based on the possibility that autoreactive B and T cells, driving myelin destruction and damage in the CNS, will be eradicated by inducing tolerance to myelin-derived peptides, without interfering with protective immunity. Several approaches involving the induction of antigen-specific tolerance have reached the clinical development phase and demonstrated promising results in phase I/II clinical trials (extensively reviewed by Willekens and Cools).[12] Given the fact that dendritic cells (DC) play a key role in controlling the immune response by steering the outcome of antigen presentation to T cells, the use of tolerance-inducing or tolerogenic dendritic cells (tolDC) may provide prospect for the treatment of MS.[16–20] Recently, we demonstrated the potential of 1α,25-dihydroxyvitamin $D_3$ to generate tolDC from MS patients.[21–23] Indeed, vitamin $D_3$-treated tolDC from MS patients display a maturation-resistant phenotype, that is, they maintain their low expression levels of costimulatory molecules

(CD80, CD83 and CD86) and anti-inflammatory cytokine profile, even following rechallenge with an inflammatory stimulus such as lipopolysaccharide. Furthermore, tolDC from MS patients are capable of inducing stable and antigen-specific T cell hyporesponsiveness. After in vitro stimulation with myelin-derived peptide-pulsed vitamin $D_3$-treated tolDC, T cells were unresponsive to the myelin peptides used, while retaining their capacity to respond to an unrelated antigen. Moreover, T cell hyporesponsiveness was robust, as T cells were not reactivated after rechallenge with immunostimulatory DC.[21–23]

Also in vivo, the administration of bone marrow-derived vitamin $D_3$-treated tolDC pulsed with $MOG_{40-55}$ has been shown to induce antigen-specific T cell tolerance in experimental autoimmune encephalomyelitis (EAE), the animal model of MS. Mice treated with $MOG_{40-55}$-pulsed bone marrow-derived vitamin $D_3$-treated tolDC before disease induction showed reduced incidence of the disease. Furthermore, when the treatment was used therapeutically in mice already showing clinical signs of the disease, the severity of the disease was significantly reduced.[24] However, repetitive injections of tolDC were necessary for a prolonged clinical effect in EAE. In this context, cryopreserved $MOG_{40-55}$-pulsed tolDC demonstrated similar clinical benefit compared with fresh tolDC, underlining the therapeutic potential and clinical applicability of cryopreserved tolDC.[21 25]

Recently, a number of phase I studies investigating the safety and feasibility of tolDC therapy for autoimmune diseases such as rheumatoid arthritis, type I diabetes or Crohn's disease were completed (table 1).[18] The first results are highly encouraging since none of the trials found safety concerns related to tolDC administration in these patients. TolDC were well tolerated and autoimmunity was not exacerbated in the patients treated.[26–29] Nevertheless, numerous questions remain, and the efficacy of antigen-specific tolDC therapy may depend on many factors, of which the route of administration is among the most important. When considering the route of delivery of DC, one needs to take into account that different routes lead to different sites of accumulation of the vaccinated DC. In most clinical studies to date, ex vivo generated DC were injected either intravenously, intradermally or subcutaneously. In humans, migration of DC towards the secondary lymph nodes is superior after intradermal injection compared with after subcutaneous injection of DC,[30–32] whereas migration of intravenously injected DC has not been monitored so far. Direct delivery of DC to lymph nodes via intranodal injection showed promising results in therapies using immunostimulatory DC in cancer,[33 34] but has not been evaluated for tolDC. It was demonstrated that brain-derived antigens can be drained from the interstitial or cerebrospinal fluid via the *lamina cribrosa* and nasal mucosa to the cervical lymph nodes, indicating that the cervical lymph nodes could be one of the first stations for the antigenic presentation at the peripheral level.[35 36] Hence, we hypothesise that intranodal injection of tolDC directly interferes with

**Table 1** Overview of clinical trials using tolDC as therapeutic intervention in autoimmune diseases

| Reference | Indication | Study design | Number of patients | Cell product and control condition | Dose | Administration mode | Primary outcome measure | Results | Immunological effects |
|---|---|---|---|---|---|---|---|---|---|
| Zubizarreta et al[44] 2019 | MS and NMO | open-label, dose-escalation, phase Ib | 8MS and 4 NMO | autologous tolDC loaded with either myelin peptides or AQP4 | $50\times10^6$, $100\times10^6$, $150\times10^6$, and $300\times10^6$ tolDC in total, separated in three independent doses administered every 2 weeks | intravenous | safety and tolerability | well tolerated no serious adverse events | ↑ IL-10 production in peptide-stimulated PBMCs and ↑ in the frequency of Tr1 |
| Bell et al[28] 2017 | Inflammatory arthritis | unblinded, dose-escalation, randomised, phase I | 9 | autologous tolDC loaded with autologous synovial fluid as a source of autoantigens | $1\times10^6$, $3\times10^6$ or $10\times10^6$ tolDC arthroscopically vs saline only | intra-articular | flare of disease in the target knee within 5 days of treatment | no target knee flares within 5 days of treatment | no consistent immunomodulatory effects in peripheral blood |
| Benham et al[27] 2015 | Rheumatoid arthritis | open-label, controlled, phase I | 34 | autologous DCs modified with a nuclear factor kappaB (NF-kappaB) inhibitor exposed to four citrullinated peptide antigens, designated "Rheumavax," | a low dose of $1\times10^6$ DCs and a high dose of $5\times10^6$ | intradermal | safety | mild adverse events | ↑ in effector T cells and an ↑ ratio of regulatory to effector T cells; ↓ in serum interleukin-15 (IL-15), IL-29, CX3CL1, and CXCL11; ↓ T cell IL-6 responses to vimentin $_{447\text{-}455}$-Cit450 relative to controls |
| Jauregui-Amezaga et al[29] 2015 | Crohn's disease | open-label, dose-escalation, phase I | 9 | autologous tolDC | first three cohorts: a single injection of $2\times10^6$, $5 \times 10^6$ or $10 \times 10^6$ tolDC; last three cohorts: 3 bi-weekly injections (same dose escalation schedule) | intraperitoneal | safety | no adverse effects | |
| Giannoukakis et al[26] 2011 | Diabetes type 1 | randomised, double-blind, phase I | 10 | autologous unmanipulated dendritic cells or tolDC | $10\times10^6$ cells once every 2 weeks for a total of four administrations | intradermal | safety | no adverse effects | ↑ in the frequency of peripheral B220+CD11c- B cells |

CXCL, Chemokine Ligand; DC, dendritic cells; IL, Interleukin; MS, multiple sclerosis; NMO, neuromyelitis optica; PBMC, Peripheral Blood Mononuclear Cell; tolDC, tolerogenic dendritic cells; Tr1, Regulatory T-cell type.

the antigen presentation and consequently, the stimulation and proliferation of autoreactive T cells. Furthermore, this route of administration omits the need for the migration requirements of the tolDC, thereby potentially enhancing the efficacy of the vaccine. Although intranodal injection is more complex, requiring ultrasound guidance, this technique is part of the daily practice at the endocrinology or radiology department of most reference hospitals.

Until now, there are no available data showing superiority of one route over others for the administration of peptide-loaded tolDC. Here, we will compare intradermal injection to intranodal injection. Only the direct comparison of these different routes of administration in two dose-escalation studies will allow us to determine if both routes are equally safe.

In conclusion, our objectives are to evaluate safety, clinical feasibility and the immunological consequences of peptide-loaded tolDC administered intranodal or intradermal in MS patients in two clinical trials. Harmonisation of the procedures for clinical, MRI as well as immune-monitoring will enable us to compare results between trials of which the study protocols will be discussed in detail here.

## METHODS AND ANALYSIS
### Study design
Two open-label, dose-escalation phase I clinical trials, MS-tolDC and TOLERVIT-MS, are designed in a coordinated and comprehensive manner and run simultaneously in Belgium (Antwerp University Hospital, Edegem) and Spain (Hospital Germans Trias i Pujol, Badalona), respectively. Patient recruitment started mid-2017 and is anticipated to end in 2020.

### Primary objectives
To evaluate the safety of administering tolDC, the occurrence and severity of adverse events (AE) will be recorded. To assess feasibility, successful production of tolDC according to good manufacturing practices (GMP) starting from a leukapheresis procedure will be assessed.

### Secondary objectives
Preliminary efficacy measures, including clinical outcomes and brain MRI, will be evaluated. In addition, whole blood lymphocyte phenotyping and cytokine profiling will be assessed before and after completion of the vaccination cycle, as well as the ability of tolDC to suppress pathogenic T cell responses. For this, myelin-specific T cell reactivity will be determined before and after completion of the vaccination cycle. All secondary objectives contribute to determination of proof of principle.

### Patient reported outcome measurements
Multiple Sclerosis Quality of Life-54 [37] will be evaluated to detect changes in general and disease-specific quality of life.

### Patient and public involvement
This study was inspired by the unmet need of finding a cure for MS, which is a key priority recognised by both patients and neurologists.[38] While patients were not involved in the design of the clinical trial, they are involved in the conduct of the study. In fact, in order to spread awareness and information about the clinical trials and its results beyond the research community with the public as a whole, a stakeholder committee was installed consisting of experts from major interest groups including members of the Flemish, Belgian and Spanish MS societies as well as Flemish and Spanish MS patients. This stakeholder committee will accompany the clinical trials from the start to the end. It has several aims: (i) to keep the stakeholder informed, (ii) to ensure that their views are considered, (iii) to challenge the project by the identification of potential emerging needs and (iv) to play an active part in the dissemination and the use of the project's results.

### Study population
#### Eligibility and enrolment
Patients with active relapsing-remitting and active progressive MS, diagnosed according to most recent McDonald criteria,[39 40] and who are not eligible for or do not want to be treated with currently available DMT, will be recruited. Patients are included after written informed consent and enrolled in the study when the inclusion and exclusion criteria are met (table 2).

#### Determination of sample size and dose-escalation procedure
The studies will be conducted according to a 'best of five' design,[41] an alternative of the traditional 3+3 design in that one additional patient is added when one or even two dose-limiting toxicities (DLT) are observed among the first three patients. Another patient is added when two DLT are observed among four treated patients. Dose escalation is allowed if DLT are observed among none of three, one of four or two of five patients, but the trial will terminate if three or more DLT are observed. A DLT is defined as a serious adverse event (SAE) that is attributable to the study cells administered, or of which the severity prevents further escalation. Dose escalation decisions will be made after all subjects in the cohort have completed at least 3 months of follow-up and when the results of the safety and tolerability analyses of the preceding dose regimen are satisfactory in the judgement of the investigators and the independent Data Safety Monitoring Board (DSMB).

An overview of the dose escalation is provided in table 3. Altogether, each phase I study is intended to accrue a total number of 9–15 evaluable patients.

### Study medication
Generation of tolDC will be carried out according to the principles and guidelines of GMP laid down in Directive 2003/94/EC. TolDC production will be performed in the GMP facilities of the Center for Cell Therapy and Regenerative Medicine of the Antwerp University Hospital (Belgium) and of the Cell Therapy Area of the Clínica Universidad de Navarra (Spain).

**Table 2** Inclusion and exclusion criteria. All the inclusion criteria must be fulfilled. The presence of any of the exclusion criteria shall exclude the patient

| Inclusion criteria | Exclusion criteria |
|---|---|
| MS according to most recent McDonald criteria | Previous use of immunosuppressive or cytostatic treatment, including mitoxantrone, cladribine, alemtuzumab or bone marrow transplantation or stem cell transplantation at any time prior to enrolment |
| Age 18–60 years | Treatment with fingolimod or natalizumab or dimethylfumarate in the past 12 weeks or teriflunomide within the past 15 weeks or ocrelizumab/rituximab within the past 6 months prior to the first administration |
| EDSS of 0–6.5 inclusive | Pregnancy or planning pregnancy in the next 12 months and breast feeding |
| First signs or symptoms at least 3 months prior to enrolment in the study | Drug or alcohol abuse |
| Active MS (relapsing and/or progressive): one relapse in the past year and/or at least one enhancing lesion on brain MRI in the past year; at least one new or enlarging T2 lesion in comparison with a reference scan from maximum 1 year before | Inability to undergo MRI assessments |
| Normal peripheral B-cell count after treatment with ocrelizumab | |
| No evidence of relapse for at least 30 days prior to start of screening and throughout during the screening phase | History of or actual signs of immunodeficiency or malignancies |
| Positive T cell reactivity response to a mix of seven myelin-derived peptides | Concurrent clinically relevant cardiac, immunological, pulmonary, neurological, renal or other major disease |
| Able to sign informed consent and comply with the protocol assessments | Active or chronic infection (hepatitis B or C, HIV, syphilis or tuberculosis) |
| No wish to be treated with currently available DMT | Splenectomy |
| Appropriate venous access and *adequate cervical lymph nodes on ultrasound mapping | |
| Use of adequate contraceptive measures. Women of childbearing potential can only be included in the study following use of adequate contraceptive measures. Accepted methods of contraception include use of hormonal contraceptives (oral, intravaginal, intrauterine or transdermal), intrauterine devices, sterilisation or postmenopausal status, use of condoms with spermicide. | |

*Only in TOLERVIT-MS.
DMT, disease-modifying treatments; EDSS, Expanded Disability Status Scale; MRI, Magnetic Resonance Imaging; MS, multiple sclerosis.

Clinical-grade autologous tolDC will be prepared from a leukapheresis. CD14$^+$ monocytes will be cultured in GMP-grade cell culture medium supplemented with 2%

**Table 3** Outline of the cell doses and patient numbers, per phase I clinical trial, in the dose escalation cohorts for intradermal and intranodal administration of tolDC

| Cohort | Treatment regimen | Patient numbers |
|---|---|---|
| 1 | 6 i.d./i.n. injections of $5 \times 10^6$ tolDC | N=3 (+1+1) |
| 2 | 6 i.d./i.n. injections of $10 \times 10^6$ tolDC | N=3 (+1+1) |
| 3 | 6 i.d./i.n. injections of $15 \times 10^6$ tolDC | N=3 (+1+1) |

N, number; i.d., intradermal; i.n., intranodal; tolDC, tolerogenic dendritic cells.

human AB serum, granulocyte macrophage colony-stimulating factor, interleukin (IL)−4 and 1α,25 dihydroxyvitamin $D_3$. At day 4, tolDC will be stimulated using an inflammatory cytokine cocktail, consisting of tumour necrosis factor-α, prostaglandin $E_2$ and IL-1β. At day 6, tolDC will be harvested, loaded with seven myelin antigens ($MBP_{13-32}$, $MBP_{111-129}$, $MBP_{154-170}$, $PLP_{139-154}$, $MOG_{1-20}$, $MOG_{35-55}$ and $MBP_{83-99}$), and cryopreserved at −196°C. Separate aliquots of the cell product will be prepared for quality control and quality assurance. This includes (i) sterility testing, (ii) cell count, (iii) viability, (iv) flow cytometric phenotyping and (v) induction of T cell hyporesponsiveness in an allogeneic mixed leucocyte reaction.

### Trial intervention: tolDC administration
In this study, six vaccine doses will be administered to each participant according to the following immunisation schedule: a total of six vaccine doses (V), with $V_{1-4}$

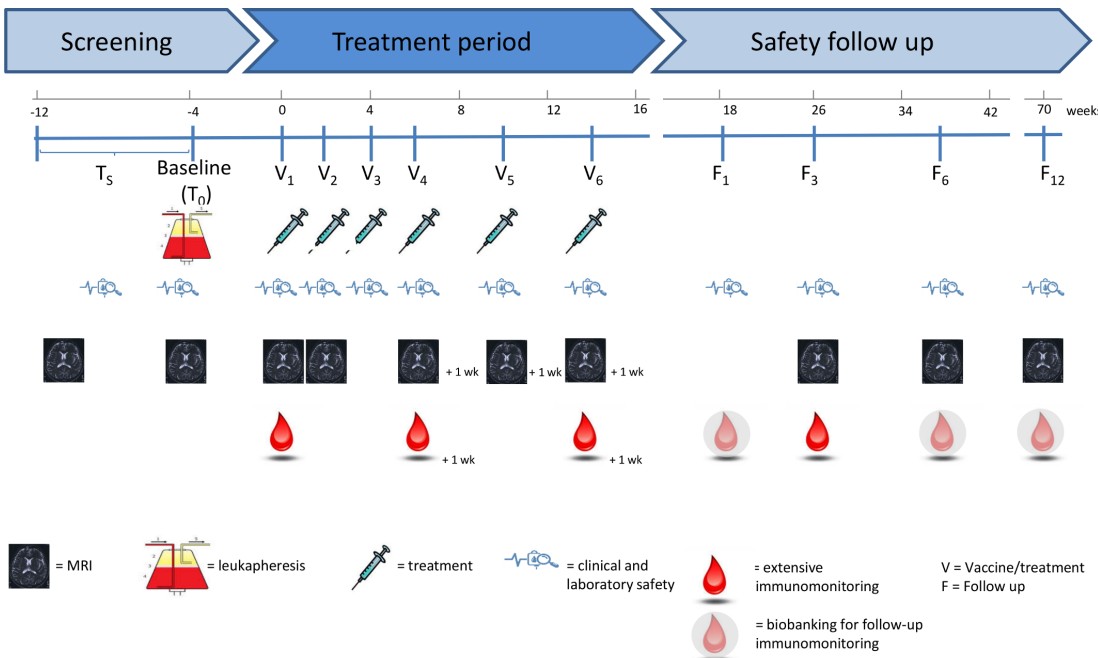

**Figure 1** Study design. Treatment will start at V₁, approximately 4 weeks after the leukapheresis. Patients will receive additional injections on week +2 (V₂), week +4 (V₃), week +6 (V₄), week +10 (V₅) and week +14 (V₆). Patients will have follow up visits one month, and 3, 6 and 12 months following the last study treatment.

at biweekly (±3 days) intervals and $V_{5-6}$ at monthly intervals (±3 days). A complete overview of the study design is depicted in figure 1. Detailed assessments per visit are shown in table 4.

In Belgium (MS-tolDC), tolDC vaccination occurs through intradermal injection at five alternating sites (100 µL/site) in the posterior neck region to ensure lymphatic drainage to superficial and deep cervical lymph nodes (5–10 cm from the cervical lymph nodes). In addition, in Spain (TOLERVIT-MS), an expert physician will inject tolDC directly in the cervical lymph nodes under echographic guidance. If the patient's anatomical conditions do not allow it (eg, small lymph nodes), the vaccine will be distributed in more nodes (max. 500 µL/node).

### Primary outcome measures
#### Safety
To evaluate the safety of administered tolDC, occurrence of AE will be recorded using clinical outcome measures, that is, physical (skin, pain and adenopathies) and neurological examination (relapses and worsening disability) and non-clinical outcome measures, that is, brain MRI, biochemical and haematological safety. The frequent MRI monitoring will allow us to monitor the safety of tolDC in MS patients objectively by measuring T1-enhancing lesions and new and/or enlarging T2 lesions. In this way, unexpected disease activity can be detected timely. The severity of AE will be defined according to the WHO toxicity grading scale. The relationship of an AE to the investigational product will be determined by the neurologists and the independent DSMB on the basis of their clinical judgement.

A proportion of the patients may experience MS relapses during the study. A relapse is defined as a new or worsening neurological symptom that occurs in the absence of fever or infection, occurs at least 30 days following the onset of a previous relapse, persists for at least 24 hours and includes an increase in Expanded Disability Status Scale (EDSS) score[42] from the previous assessment matching one of the following: (i) an increase of ≥1 on the total scale score; or (ii) an increase of ≥2 points on one of the appropriate Functional System Scores (FSS) or (iii) an increase of >1 point on two or more of the appropriate FSS. Patients will be instructed to notify their neurologist as soon as possible but within 72 hours the latest and will be examined within 7 days of onset of the symptoms. In case of an MS relapse, a course of high-dose steroids can be administered at the discretion of the treating neurologist.

Study treatment must be discontinued for a given patient if the investigator determines that continuing would result in a significant risk for that patient. In case of terminating the study treatment, patients will still be monitored for safety issues for the duration of the study.

#### Feasibility
To evaluate the feasibility, successful production of tolDC after leukapheresis will be evaluated. This includes the production of sufficient numbers of tolDC and compliance of the final investigational product to the prespecified release criteria following cryopreservation.

### Secondary outcome measures
#### Clinical evaluation
The following clinical and laboratory assessments will be performed during a complete physical examination:

**Table 4** Study calendar. The different visits, examinations, tests and the tolDC administration are detailed on the tabular study schedule overview.

| | Screening | Baseline | Treatment period | | | | | | | | Safety follow-up | | | |
| | Ts | (T0) | V1 | V1 +1day | V2 | V3 | V4 | V5 | V6 | F1 | F3 | F6 | F12 |
|---|---|---|---|---|---|---|---|---|---|---|---|---|---|
| Informed consent | • | | | | | | | | | | | | |
| Inclusion and exclusion criteria | • | | | | | | | | | | | | |
| IFN-γ EliSPOT (T cell reactivity assay) | • | | • | | | | •+1w | | •+1w | | • | | |
| Leukapheresis | | • | | | | | | | | | | | |
| TolDC preparation | | • | | | | | | | | | | | |
| TolDC treatment | | | • | | • | • | • | • | • | | | | |
| Patient evaluation: | | | | | | | | | | | | | |
| Vital signs (HR, BP, …) | • | | • | | • | • | • | • | • | • | • | • | • |
| ECG | • | | • | | • | • | • | • | • | • | • | • | • |
| Blood analysis for safety (1×5 mL EDTA tube and 2×10 mL serum tubes) | • | | • | | • | • | • | • | • | • | • | • | • |
| Urine pregnancy test | • | | • | | • | • | • | • | • | • | • | • | • |
| Full neurological examination (EDSS, 9-HPT, T25FW, SDMT, MSQOL-54) | • | | • | • | | | | | | | | | |
| Pain score after treatment injection | | | • | | • | • | • | • | • | | | | |
| MRI | • | | • | | | | •+1w | •+1w | •+1w | • | • | • | • |
| Immunomonitoring (10×10 mL heparin tubes+1 serum tube) | | | • | | | | •+1w | | •+1w | | • | | |
| Biobanking for follow-up immunomonitoring | | | | | | | | | | | | | |
| (Serious) Adverse events and concomitant medication | Continuous | | | | | | | | | | | | |

BP, Blood Pressure; ECG, Electrocardiogram; EDSS, Expanded Disability Status Scale; 9-HPT, 9 Hole Peg Test; MRI, Magnetic Resonance Imaging; MSQOL-54, Multiple Sclerosis Quality of Life-54; SDMT, Symbol Digit Modalities Test; T25FW, Timed 25 Foot Walk; tolDC, tolerogenic dendritic cells.

vital signs, ECG, routine blood and urine samples, urine pregnancy test in female patients of reproductive potential and serological screening tests, according to the study schedule (figure 1 and table 4). Concurrent drug use or new AE will be reviewed and recorded.

EDSS is based on a standardised neurological examination and measures impairments in eight functional systems, including vision, brainstem, pyramidal, cerebellar, sensory, bowel and bladder, mental (cerebral) and ambulation . It has been used for over 20 years as a clinical outcome measure of MS disease progression and consists of a 10-point scale of disease severity ranging from 0, that is, no disability, to 10, that is, death from MS.[42] All neurologists involved in patient evaluation will be EDSS training certified.

EDSS will be supplemented by three well-known, quantitative, continuous tests that evaluate ambulation (walking speed by the timed 25 foot walk test or T25FW, arm dexterity and function by the 9 Hole Peg Test or 9-HPT, and cognition by the Symbol Digit Modalities Test or SDMT).

The effect on disability progression will be characterised by reporting the proportion of patients who are free from disease progression. For this, disability will be assessed based on a sustained clinically relevant change seen in any one of the disability assessments: EDSS, T25FW, SDMT or the 9-HTP. Disease progression is defined by: 3-month sustained increase from baseline in the EDSS score (1 point in patients with baseline EDSS score 0 to 5.0; 0.5 point in patients with baseline EDSS score of 5.5–6.5) or 3-month sustained increase of at least 20% from baseline in the time taken to complete the T25FW or 3-month sustained increase of at least 20% from baseline in the time taken to complete the 9-HPT. Percent change from baseline after treatment in SDMT score will also be recorded.

### MRI acquisition and evaluation

MRI is the current gold standard for non-clinical monitoring of MS, and MRI-derived markers have been established as standard outcome measures to monitor the treatment response in various MS clinical trials. MRI will be performed on the same 3T scanner throughout the study. The MRI protocol includes a 3PLANE scout, 3D T1-weighted image pregadolinium and postgadolinium (Dotarem, 20 mL) administration (voxel resolution 0.9×0.9×0.9, TR2300.0, TE2.29, TI 900.0) and a 3D FLAIR image (voxel resolution 0.4×0.4×0.9, TR5000.0, TE387.0, TI1800.0). The quantification of number of T2 lesions, T2 lesion load, number and volume of Gd-enhancing lesions, based on 3D FLAIR images and 3D T1-weighted images, is completely automatic avoiding inter-rater or intra-rater variability. The lesions are quantified within different brain regions. Disease activity or progression on MRI will be evaluated by pretreatment versus ongoing and post-treatment: (i) change in mean number of enhancing lesions; (ii) change in number of T1 Gd-enhancing and/or new or enlarging T2 lesions; (iii) percent change in T2 lesion load and (iv) percent change in brain volume. In order to compare MRI outcome measures from different studies, a standardised acquisition is guaranteed by (i) a uniform MRI protocol, (ii) the use of standard operating procedures for image acquisition and upload and (iii) training of local MRI operators and study coordinators for acquisition and upload.

### Immune-monitoring

To evaluate therapy-related changes in the immune cell profile, peripheral blood (10×10 mL heparin tubes) will be sampled according to the time points depicted in figure 1 and table 3, and analysed by multiparameter flow cytometry prior to, during and after vaccination. In an attempt to cover the main leucocyte subsets of peripheral blood, the following subsets and their activation status will be enumerated in whole blood samples: CD4$^+$ and CD8$^+$ T cell subpopulations, B cell subsets, natural killer (NK) cells, NKT cells and myeloid cells. Myelin-specific T cell reactivity will be determined before, during and after completion of administration of tolDC. For this, responsiveness of T cells to myelin antigens will be investigated. As a control, the responsiveness of T cells to unrelated antigens, for example, cytomegalovirus or tetanus toxoid, will be addressed. In doing so, we will be able to assess ex vivo the potential risk of inducing opportunistic infections by administering tolDC to MS patients. Finally, patient materials from the studies will be specifically biobanked. We envisage to maximally performing batch measurements and centralised immune monitoring analysis as soon as a dose cohort has reached 3 months follow-up of all patients. For this, extensive immune cell profiling as well as cytokine production of T cells will be analysed by multiparameter flow cytometry. Using lineage-specific as well as activation markers, proportions and activation status of the different immune cell subsets will be determined. Besides, antibody titers and memory B cell analysis will be performed.

### Patient reported outcome measurements

MS QOL-54[37] will be evaluated to detect changes in general and disease-specific quality of life.

### Data management and monitoring

Adequate and accurate patient records will be kept enabling the appropriate and required documentation of the study and subsequent verification of the collected data. All data are completed in the electronic case report form for each patient enrolled in this study, including patients who did not start with the investigational treatment. A central data manager affiliated to a clinical research organisation will perform source data verification. In doing so, study compliance will be monitored, thereby assuring the protection of the rights, safety and well-being of study subjects.

### Analysis

Given the design of the study (phase I) and its specific primary end point (safety), no confirmatory statistical

testing will be performed. P-values will be calculated but interpreted with caution. Study investigations will be exploratory and conclusions will be based on the complete set of patient evidence.

Analyses of safety variables will be performed with the safety population (all the patients treated with at least one dose of the cell product) by available data only analysis. For demographic and safety analysis, that is, the incidence of AE, laboratory values and changes in vital signs, descriptive statistics will be calculated including frequencies for categorical variables and mean and SD, or median and IQR for quantitative parameters. In brief, EDSS and other repeatedly measured variables will be analysed by means of Mixed Models for Repeated Measurements. In case model assumptions are violated (eg, non-Gaussian errors) or when dealing with ordinal efficacy variables, alternative methods (including non-parametric methods and ordinal regression methods for repeated measurements) will be used. The percentage of patients with 1-point in EDSS improvement will be estimated using a binomial regression model including the treatment and the baseline EDSS. The remainder of variables will be analysed according to the appropriate statistical test: $\chi^2$ or Fisher's exact test to compare categorical variables, the dependent or independent t-test for continuous Gaussian-distributed variables and the Wilcoxon or Mann-Whitney test for ordinal and non-Gaussian continuous data. The significance level alpha will be set at 0.05 for two-tailed analysis. Variables to assess the analysis of secondary and tertiary outcome measures will be performed with per protocol population. Continuous variables as a minimum will be described by number of total and non-missing observations (n) and the appropriate location-scale statistics including arithmetic mean, SD, minimum, median, Q1–Q3 and maximum. Categorical variables will be presented using the number of non-missing observations (n) or the number of patients in the population (N) as applicable and percentages (%). Two-sided 95% (exact) CI will be provided when relevant.

### Patient protection

The study will be conducted in agreement with either the Declaration of Helsinki or the laws and regulations of the country, whichever provides the greatest protection of the patient. In the Belgian context, the Law of 7 May 2004 ('Wet van 7 mei 2004 inzake experimenten op de menselijke persoon') applies. The study will be conducted in agreement with the ICH Harmonised Tripartite Guideline for Good Clinical Practice.

### Informed consent

All patients will be informed of the aim of the study, the possible AE, the procedures and possible hazards to which he/she will be exposed. They will be informed as to the strict confidentiality of their patient data, but that their medical records may be reviewed for trial purposes by authorised individuals other than their treating physician.

### Dissemination

The results of the clinical trials will be published in a peer-reviewed journal. In addition, the results will be presented at scientific conferences and to patient associations. In particular for Spanish patient associations, dissemination of results will be handled by the Spanish Agency of Medicines and Medical Devices whose content is written in layman's terms. On completion of the trial and after publication of the study results, data requests can be submitted to the researchers.

## DISCUSSION

Although the first phase I clinical trials have demonstrated promising results with regard to safety of administering tolDC in other autoimmune diseases, numerous questions remain concerning which dose, treatment schedule or route of administration is best with regard to safety, efficacy and related costs of treatment with tolDC. In this collaborative effort, each patient will receive six repetitive injections of 5, 10 or $15 \times 10^6$ autologous myelin-derived peptide mix-loaded tolDC, intradermal or intranodal, that is, four administrations once every 2 weeks and two administrations once every 4 weeks. Previously, others evaluated the safety of intravenous administration of tolerogenic DC in MS and neuromyelitis optica patients (table 1).[43 44] In the current study, harmonisation of clinical, MRI and immunological evaluations of the patients will enable us to compare results between two phase I clinical trials evaluating the safety and feasibility of autologous tolDC administration in patients with active MS. To our knowledge, this will be the first time that different routes of administration are set side by side for a cell therapy product.

In conclusion, our protocols envisage to restore tolerance to predefined myelin-peptide antigens using peptide-loaded tolDC. From the results of these two phase I clinical studies, the optimal dose and administration route will be selected for future phase II trials investigating the efficacy of this patient-tailored treatment in MS.

**Author affiliations**
[1]Department of Neurology, University Hospital Antwerp, Edegem, Belgium
[2]Laboratory of Experimental Hematology, Vaccine & Infectious Disease Institute (VAXINFECTIO), University of Antwerp Faculty of Medicine and Health Sciences, Wilrijk, Belgium
[3]Multiple Sclerosis Unit, Department of Neurosciences, Hospital Universitari Germans Trias i Pujol, Badalona, Spain
[4]Department of Medicine, Universitat Autònoma de Barcelona, Cerdanyola del Vallés, Spain
[5]Division of Immunology, LCMN, Hospital Universitario Germans Trias i Pujol and Research Institute, Badalona, Spain
[6]Department of Cellular Biology, Physiology and Immunology, Universitat Autònoma de Barcelona, Bellaterra, Spain
[7]Center for Cell Therapy and Regenerative Medicine, University Hospital Antwerp, Edegem, Belgium
[8]Department of Radiology, University Hospital Antwerp, Edegem, Belgium
[9]Icometrix NV, Leuven, Belgium
[10]C-Clear Partners, Mortsel, Belgium
[11]Department of Nephrology, University Hospital Antwerp, Edegem, Belgium
[12]Haematology-Cell Therapy Area, clinica universidad de navarra, Pamplona, Spain

[13]Immunology and Immunotherapy Department, Clinica Universidad de Navarra, Pamplona, Spain

[14]Program of Haematology-Oncology, CIMA, Universidad de Navarra, Pamplona, Spain

[15]Lygature, Utrecht, The Netherlands

[16]Department of Neurology, University Hospital Munster, Munster, Germany

[17]Department of Neurology, University of Munster, Munster, Germany

[18]Department of Immunopathology, Sanquin Research, Amsterdam, The Netherlands

[19]Landsteiner Laboratory, Academic Medical Centre, University of Amsterdam, Amsterdam, The Netherlands

[20]Clinical Research Polyvalent Unit, Clinial Trial Unit-Spanish Clinical Research Network, Germans Trias i Pujol Health Sciences Research Institute, Badalona, Spain

[21]Department of Pharmacology, Therapeutic and Toxicology, Universitat Autònoma de Barcelona, Cerdanyola del Vallès, Spain

[22]Institut de Diagnòstic per la Imatge, Hospital Universitari Germans Trias i Pujol, Badalona, Spain

[23]Interuniversity Institute for Biostatistics and statistical Bioinformatics (I-BioStat), Universiteit Hasselt, Hasselt, Belgium

[24]Centre for Health Economic Research and Modelling Infectious Diseases, Vaccine & Infectious Disease Institute (VAXINFECTIO) & Center for Statistics, University of Antwerp Faculty of Medicine and Health Sciences, Wilrijk, Belgium

**Acknowledgements** We thank all the patients who contributed to the preclinical research by donating samples. Our gratitude goes to Marleen Breuls, MS nurse and to the study nurses of the Neurology Department of the Antwerp University Hospital, Linda Wagemaekers, Caroline Vinck and Maren Wyckmans as well as Sandra Vidal, study nurse at Clinical Trial Unit in Research Foundation Germans Trias i Pujol-IGTP. We thank the Spanish Clinical Research Network-SCReN for the support on Pharmacovigilance and Case Report Form implementation for the study. We thank also Jorge Luis Reverter, at Endocrinology Department at Hospital Germans Trias i Pujol for the treatment administration. We thank the patients (MDB and PAM) and Belgian and Spanish MS patient organisations (Christiane Tihon, Luc de Groote, Leyre Avellanal) as well as the experts (Ed Geissler, Antonio Ucelli, David Wraith, Catharien Hilkens, Tom Bosschaerts) for their participation in the stakeholder committee. We thank the external scientific advisory board (Ed Geissler, Antonio Ucelli, David Wraith, Catharien Hilkens) and the data safety monitoring board members (Christian Sindic, Ludo Vanopdenbosch, Joan Albert Arnaiz) for their contributions.

**Collaborators** RESTORE Consortium collaborators: Naomi Ooms (Center for Cell Therapy and Regenerative Medicine, Antwerp, Belgium); Dirk Smeets (Icometrix, Leuven, Belgium), Leone Bock, Denis Groot, Wim-Jan Koot, Janwillem Boiten, Jorg Janssen, Sjaak Peelen (Lygature, Utrecht, The Netherlands); A. Turksma, R. Vanlier, T. Rispens, D. DiBlasi, Iris Claessen (Sanquin, Amsterdam, The Netherlands); M. Puig-Domingo, Laia Lagunas Vila, David Basanta Pons (Germans Trias y Pujol Research Institute, Badalona, Spain); Itziar Astiasarán, Javier Mata Rodríguez (University of Navarra, Spain); Antje Albring, Jana Arnholdt (University of Munster, Germany)

**Contributors** BW, ZB, CRT, EMC, WVH and NC conceived and designed the study. WPL, GN, MDL, GA, AR, MC, IW and NC participated in logistical planning of the study. BW and SPR wrote the initial draft of the manuscript. BW, SPR, CRT, MJM, IW, EMC and NC wrote and reviewed the manuscript. NH provided the statistical support for the sample size estimates and the design of the statistical analysis. BW, SPR, MJM, JD, WPL, GN, MDL, IW, PC, PP, WVH, AR, TB, GA, MC, JNB, ATS, BQS, ALDC, SI, FP, AK, HV, CCG, HW, SMVH, ATB, AMB, AMV, NH, ZB, EMC, NC and CRT made significant contributions to the development and conceptualisation of the protocol, reviewed the draft versions of this paper and have read and approved the final manuscript.

**Funding** This work was supported by positive discussion through the A FACTT network (Cost Action BM1305: www.afactt.eu). COST is supported by the EU Framework Program Horizon 2020. This RESTORE project has received funding from the European Union's Horizon 2020 research and innovation program under grant agreement number 779316. Further support was provided by an applied biomedical research project of the Institute for the Promotion of Innovation by Science and Technology in Flanders (IWT-TBM 140191), by projects PI11/02416, PI14/01175,PI16/01737 and PT13/0002/0038 (Platform for Clinical Research and Clinical Trial Units, Spanish Clinical Research Network, SCReN), integrated in the Plan Nacional de I+D+I and co-supported by the Health Institute Carlos III - Subdirección General de Evaluación y Fomento de la Investigación de la Spanish Ministry of Economy and Competitiveness and the Fondo Europeo de Desarrollo Regional (FEDER), and project 07/2410 Fundació La Marato de TV3. Furthermore, the authors received research funding from Sanofi Genzyme,

Belgium. Judith Derdelinckx holds a PhD fellowship from the Research Foundation Flanders (FWO). Dr Presas-Rodríguez is a neurologist who has received a grant of Hospital Germans Trias i Pujol ('Germans Trias Talents 2016-2018') to work on this project. Dr Willekens is a neurologist at the Antwerp University Hospital supported by a research fellowship (2016-2018) of the University of Antwerp to work on this project and she currently holds a clinical PhD fellowship from the Research Foundation Flanders (FWO 1701919N). Spanish Patient association 'Treball de Vida' (Associació d'Afectats d'Esclerosi Múltiple del Barcelonès Nord i Maresme) and patient Ana Mª Calvo Marsal have donated funding to the Hospital Germans Trias i Pujol MS Unit.

**Competing interests** CCG received speaker honoraria and travel expenses for attending meeting from Genzyme, Novartis Pharma GmbH, and Bayer Health Care. Her work is funded by the German Ministry for Education and Research (BMBF; 01GI1603A) and the German Research Foundation (DFG; GR3946/3-1 and SFB128 A09). HW receives honoraria for acting as a member of Scientific Advisory Boards and as consultant for Biogen, Evgen, MedDay Pharmaceuticals, Merck Serono, Novartis, Roche Pharma AG, Sanofi-Genzyme, as well as speaker honoraria and travel support from Alexion, Biogen, Cognomed, F. Hoffmann-La Roche Ltd., Gemeinnützige Hertie-Stiftung, Merck Serono, Novartis, Roche Pharma AG, Sanofi-Genzyme, TEVA, and WebMD Global. Prof. Wiendl is acting as a paid consultant for Abbvie, Actelion, Biogen, IGES, Novartis, Roche, Sanofi-Genzyme, and the Swiss Multiple Sclerosis Society. His research is funded by the German Ministry for Education and Research (BMBF; 01FI1601E, 01GI1603A, and 01GI1603D), Deutsche Forschungsgesellschaft (DFG; SFB128 A09, A10, Z02, V and SFB1009 A03), Else Kröner Fresenius Foundation, Fresenius Foundation, Hertie Foundation, NRW Ministry of Education and Research, Interdisciplinary Center for Clinical Studies (IZKF) Muenster and RE Children's Foundation, Biogen GmbH, GlaxoSmithKline GmbH, Roche Pharma AG, Sanofi-Genzyme. The institution of BW receives honoraria for acting as a member of Scientific Advisory Boards for Biogen, Merck Serono, Roche, Sanofi-Genzyme, Novartis and speaker honoraria and travel support from Biogen, Merck Serono, Roche, Sanofi-Genzyme, Novartis, TEVA. CRT receives honoraria for acting as a member of Scientific Advisory Boards for Biogen, and Merck Serono, and speaker honoraria or travel support from Biogen, Merck Serono, Roche, Sanofi-Genzyme and Novartis. SPR receives speaker honoraria or travel support from Biogen, Merck Serono, Roche, Sanofi-Genzyme and Novartis. PP is a medical advisory board member of Icometrix NV. The other authors report no conflict of interest.

**Patient and public involvement statement** See methods and analysis section

**Patient consent for publication** Not required.

**Ethics approval** Ethics approval was obtained for both phase I clinical trials. The study protocol was approved by the Federal Agency for Medicines and Health Products (FAMHP, Belgium), the Spanish Agency of Medicines and Medical Devices (Spain), the Ethics Committee of the Antwerp University Hospital (Belgium) and the Ethics Committees of Hospital Germans Trias i Pujol and Clínica Universidad de Navarra.

**Provenance and peer review** Not commissioned; externally peer reviewed.

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
