## [Reviewer comments · BMJ Open]

ARTICLE DETAILS

TITLE (PROVISIONAL)	Tolerogenic dendritic cell-based treatment for multiple sclerosis (MS): a harmonized study protocol for two phase I clinical trials comparing intradermal and intranodal cell administration
AUTHORS	Willekens, Barbara; Presas-Rodríguez, Silvia; Mansilla, MJ; Derdelinckx, Judith; Lee, Wai-Ping; Nijs, Griet; De Laere, Maxime; Wens, Inez; Cras, Patrick; Parizel, Paul; Van Hecke, Wim; Ribbens, Annemie; Billiet, Thibo; Adams, Geert; Couttenye, marie-madeleine; Navarro-Barriuso, Juan; Teniente-Serra, Aina; Quirant-Sánchez, Bibiana; Lopez-Diaz de Cerio, A; Inogés, S; Prosper, Felipe; Kip, Anke; Verheij, Herman; Gross, CC; Wiendl, Heinz; Van Ham, SM; Ten Brinke, Anja; Barriocanal, Ana María; Massuet-Vilamajó, A; Hens, Niel; Berneman, Zwi; Martínez-Cáceres, Eva; Cools, Nathalie; Ramo-Tello, Cristina

VERSION 1 - REVIEW

REVIEWER	Jeremy Chataway University College London (UCL) London UK
REVIEW RETURNED	24-Apr-2019

GENERAL COMMENTS	This is the protocol for a novel tolerance inducing approach in multiple sclerosis (MS). The design is for two open-label, dose-escalation, phase 1 studies which will be coordinated together. The studies commenced in 2017 The protocol is clearly described and has ethical approval.
---

REVIEWER	Richard Nicholas Imperial College London, UK non-financial support from Roche, personal fees and non-financial support from Novartis, personal fees and non-financial support from Biogen, grants from UK MS Society
REVIEW RETURNED	07-May-2019

GENERAL COMMENTS	Thank-you asking me to review this work. This is an interesting approach and certainly deserves to be pursued. It is a complex field with many unsuccessful approaches in the past and it is essential to clarify why this approach is building on those studies.
---

	The abstract and introduction needs some revision. The views regarding long term prognosis is not shared by all (see recent BMJ views) and should be toned down eg remove extraordinary. If treatment was so good why do we need new therapies targeting relapse. We in fact need therapies that address progression and given this is a phase 1 trial this is a long way off. There have been a number of prominent failures in treatments when addressing new areas of the immune system based on good in vitro evidence. Even vitamin therapies have been associated with increasing relapses. The recent withdrawal of a MS therapy stimulating the immune system does not necessarily mean there will be less impact from this approach. It would be helpful to focus a little more on why the nodal approach is being used and its acceptability. This would have more issues for subjects undergoing therapy especially as it requires echographic guidance. It could be associated with a number of local issues and it may have been helpful to have patient feedback on the acceptability of this approach It would be helpful to have a table summarising the prior studies and their outcomes to get an idea of the range of its prior use. Strengths and limitations: I do not think no blinding is limitation of a phase 1 study. Methods and Analysis Targeting those with active MS/active progressive who turned down therapy will make this a more benign group. This problem is an issue for all studies but may be a limitation.
--	--

VERSION 1 – AUTHOR RESPONSE

Reviewer 1, dr. Jeremy Chataway.

This is the protocol for a novel tolerance inducing approach in multiple sclerosis (MS). The design is for two open-label, dose-escalation, phase 1 studies which will be coordinated together. The studies commenced in 2017 The protocol is clearly described and has ethical approval.

We thank the reviewer for taking the time to review our manuscript.

Reviewer 2, dr. Richard Nicholas.

Thank-you asking me to review this work. This is an interesting approach and certainly deserves to be pursued.

We thank the reviewer for his review and useful comments. We have adapted the manuscript accordingly.

It is a complex field with many unsuccessful approaches in the past and it is essential to clarify why this approach is building on those studies.

Thank you for this comment. This topic has been extensively reviewed in a recent paper by Willekens and Cools (CNS drugs 2018) and we kindly refer the readers to this open access paper.

The abstract and introduction needs some revision.

The views regarding long term prognosis is not shared by all (see recent BMJ views) and should be toned down eg remove extraordinary.

We removed the word 'extraordinary' and toned down the efficacy of current DMT.

If treatment was so good why do we need new therapies targeting relapse.

We are still in need of new relapse-targeting therapies with good safety profiles that can lead to long-lasting effects and that are more selective than current therapies. This reasoning is stated in the introduction.

We in fact need therapies that address progression and given this is a phase 1 trial this is a long way off.

We fully agree that a therapy that tackles progression is another unmet need. However, this is not the scope of this manuscript.

There have been a number of prominent failures in treatments when addressing new areas of the immune system based on good in vitro evidence. Even vitamin therapies have been associated with increasing relapses. The recent withdrawal of a MS therapy stimulating the immune system does not necessarily mean there will be less impact from this approach.

Recently, an overview of current therapeutic vaccination strategies discussing failures, successes and future directions of these approaches by Willekens & Cools was published in CNS drugs 2018. In the two open-label, dose-escalation, phase I studies described here, we do not anticipate paradoxical disease worsening due to the toIDC therapy. Nonetheless, we will perform frequent neurological and brain MRI assessments to detect any safety issue as early as possible.

It would be helpful to focus a little more on why the nodal approach is being used and its acceptability. This would have more issues for subjects undergoing therapy especially as it requires echographic guidance. It could be associated with a number of local issues and it may have been helpful to have patient feedback on the acceptability of this approach.

We thank the reviewer for this suggestion and accordingly added a paragraph discussing this topic in the manuscript (see page 8, manuscript with track changes).

It would be helpful to have a table summarising the prior studies and their outcomes to get an idea of the range of its prior use.

We have added a table with a summary of prior studies using toIDC treatment in autoimmune diseases.

Strengths and limitations: I do not think no blinding is limitation of a phase 1 study.

We have removed this sentence.

Methods and Analysis

Targeting those with active MS/active progressive who turned down therapy will make this a more benign group. This problem is an issue for all studies but may be a limitation.

This is indeed possible, however we have also screened patients with bad prognostic factors as well as early MS patients who have a belief in this approach.

VERSION 2 – REVIEW

REVIEWER	Richard Nicholas Imperial College London Biogen, Novartis, Roche honoraria for speaking, advisory boards and participating in clinical research
REVIEW RETURNED	21-Jul-2019

GENERAL COMMENTS	Thank-you for your revisions they have clarified the paper for me
---